# A New Robotic Endoscope Holder for Ear and Sinus Surgery with an Integrated Safety Device

**DOI:** 10.3390/s22145175

**Published:** 2022-07-11

**Authors:** Guillaume Michel, Philippe Bordure, Damien Chablat

**Affiliations:** 1CHU Nantes—Centre Hospitalier Universitaire de Nantes, 44000 Nantes, France; guillaume.michel@chu-nantes.fr (G.M.); philippe.bordure@univ-nantes.fr (P.B.); 2École Centrale Nantes, Nantes Université, IMT Atlantique, CNRS, INRIA, LS2N, UMR 6004, 44300 Nantes, France

**Keywords:** surgery, endoscope, RCM, safety, sinus, ear, parallel robot, spring energy

## Abstract

In the field of sinus and ear surgery, and more generally in microsurgery, the surgeon is faced with several challenges. The operations are traditionally carried out under binocular loupes, which allows for the surgeon to use both hands for a microinstrument and an aspiration tool. More recently, the development of endoscopic otological surgery allowed for seeing areas that are difficult to access. However, the need to handle the endoscope reduces the surgeon’s ability to use only one instrument at a time. Thus, despite anaesthesia, patient motions during surgery can be very risky and are not that rare. Because the insertion zone in the middle ear or in the sinus cavity is very small, the mobility of the endoscope is limited to a rotation around a virtual point and a translation for the insertion of the camera. A mechanism with remote center motion (RCM) is a good candidate to achieve this movement and allow for the surgeon to access the ear or sinus. Since only the translational motion along the main insertion axis is enabled, the ejection motion along the same axis is safe for the patient. A specific mechanism allows for inserting and ejecting the endoscope. In a sense, the position is controlled, and the velocity is limited. In the opposite sense, the energy stored in the spring allows for very quick ejection if the patient moves. A prototype robot is presented using these new concepts. Commercially available components are used to enable initial tests to be carried out on synthetic bones to validate the mobility of the robot and its safety functions.

## 1. Endoscope Holders and General Anaesthesia

In otologic surgery, surgeons are increasingly using endoscopes instead of the traditional microscopic approach (Figure 1a). The benefit of using an endoscope for otologic surgery is to improve the visualization of the middle ear without the need for a retroauricular incision or mastoidectomy [1]. The use of the endoscope has shown its value in detecting recurrences of cholesteatoma in areas that are difficult to access [2].

In sinus surgery, the endonasal approach using endoscopes has been the standard technique for 30 years. Numerous studies demonstrated the value of endonasal surgery, with lower complication rates compared to external surgery [3] and better quality of life postoperatively [4].

When the surgeon holds the endoscope, the surgery is performed with a single hand (Figure 1b). This one-handed surgery is responsible for a long learning curve, as it is used in a very restricted workspace, with an average volume of the eardrum of 0.99 cm^3^ [5]. The surgeon should not touch the sensitive components of the middle ear (ossicles, facial nerve, …). With only one hand to operate, they may have major difficulties in achieving haemostasis when bleeding occurs in otologic surgery [1].

Although the working space is larger in the sinus than that in the ear [5], progress is still needed in ergonomics during endoscopic surgery. The analysis of operating videos revealed that 20–50% of surgical time is spent on tasks such as blood suctioning to allow for the operating area to be seen [6]. Another difficulty is the impossibility to stretch a tissue with one hand before cutting it. Most of these limitations could be improved by allowing the surgeon to work with both hands rather than just one and presumably decrease the operating time.

A solution offered to the surgeon to solve this problem is a mechanism to handle the endoscope to facilitate two-handed surgery. Endoscope holders are now commercialised for ear surgery (Robotol) or sinus surgery (Endofix Exo) [7].

However, these devices have no integrated safety device, but in middle-ear surgery, the endoscope is placed within millimetres of critical structures such as the ossicular chain or facial nerve [8]. For example, when a patient involuntarily moves their head under general anaesthesia and suddenly moves against a fixed endoscope, this may lead to heat and mechanical damage to the structures of the middle ear [9]. This can lead, for example, to facial palsy or hearing loss.

Thus, movements during surgery can be very risky for the patient and are not that uncommon.

There are few data on patient movement during ear surgery, although experience shows that this happens regularly (coughing, involuntary movements, early waking). However, the movement of a few millimetres against a rigid endoscope could have serious consequences. In a retrospective study of 100 consecutive patients undergoing otologic procedures [10], there was one instance where surgery was temporarily interrupted due to patient movement. However, in a recent prospective study [8], head motion was observed in 40% of cases during ear surgeries. These values were close to those found in other prospective studies: a comparison of different sedation protocols (propofol–fentanyl and midazolam–fentanyl) revealed 30 to 35% of movements during middle-ear surgeries [11]; another comparative study found 23% (remifentanil-based anaesthesia) to 65% (propofol) of movements during surgery [12]. During robotic surgery, the maintenance of a deep neuromuscular blockade should be considered to improve safety by preventing patient movement [13]; but these drugs prevent the monitoring of the facial nerve, which is often required in otologic surgery.

Berges et al. [8] measured these movements during otologic surgery: head motion in 40% of cases with a maximal linear acceleration of 0.75 m/s^2^ and angular velocity of 12.50 degrees/s. In their opinion, these findings legitimised concerns that static endoscope holders represent a significant surgical risk, and demonstrated the need for a dynamic holder that could react to unintended head motion.

According to this study, a dynamic endoscope operator should react in less than 0.2 s to prevent damage to the middle ear [8].

On the basis of these data, static endoscope holders could represent a surgical risk, and justify the need to integrate safety devices in robotic requirements.

## 2. Analysis of the Need for the Mobility of an Endoscope Holder

Numerous projects from research laboratories and companies have created robots to assist the surgeon during operations. However, there is no robot that can perform operations in the ears and sinuses, and the same surgeon performs these operations [7].

An endoscope robot should enable the surgeon to see particular areas of the middle ear and sinuses. These regions were characterised in [5] to define the rotational movements of the endoscope, taking into account the anatomy of the ears and sinuses, and the mobility of the patient when lying on the operating table. This study allows for the characterisation of the rotational movements of the RCM. In order to position the robot relative to the patient, the translational movements of its base must be added. This last part is not dealt with in this article because a simple mechanism with three degrees of freedom can be used.

The mobility of the endoscope inside the ear and sinuses is, therefore, mainly a rotational motion around a fixed point. In the case of using a straight optic, rotation around the insertion axis is not necessary: two actuated rotations are necessary. Thus, an additional translation movement along the axis of the endoscope is necessary for two reasons:Insert the endoscope into the port and withdraw it for cleaning if blood is on the optic. This movement is controlled to allow the surgeon to resume activity after cleaning quickly.Quickly removes the endoscope in case of patient movement to ensure patient safety.

For existing robots, the translational movement is performed manually with the Endoscope Robot (Medineering) [14], with the complete movement of the arm with the Endofix Exo [15], or with an actuator located at the end effector of the robot, as for the Da Vinci Robot [16] (the size of the Da Vinci robot endoscope does not allow for operations in the ears). In this case, the speed of movement is limited by the power of the motor that is supported by the arm.

## 3. Mechanism Solution for Patient Safety

### 3.1. Patent Description

The proposed patient safety solution is derived from a patent [17]. Figure 2 and Figure 3 schematically illustrate a robot or otological surgical aid with an insertion–extraction system for a tool such as an endoscope. The insertion–extraction system is in such a position that the tool is lowered and located in a workspace area. Robot comprises a base, an arm, the insertion–extraction system, and a tool. The robot is shown in an (Oxyz) reference frame, with point O being located at the intersection of axes Ox and Oz, and another axis (Oy) is perpendicular to the (Oxz) plane. The workspace area is located near the intersection between an insertion axis and the Ox axis.

The arm of the robot comprises a deformable double parallelogram consisting of a first parallelogram comprising rods 1–4, and connected to a second parallelogram comprising Rods 3–6. Vertical rear rod 3 and horizontal intermediate rod 4 are common to both parallelograms. Rods 1 to 6 are connected to each other by revolute joints A to G, allowing for the rotations of axes parallel to axis Oy, and allowing in particular the rotation of axis Oy between lower rod 1 and rear rod 3. The insertion axis of the tool is oriented in the direction of front rod 5, parallel to rear rod 3 and vertical intermediate rod 2.

The arm of the robot is connected to the base with an actuation mechanism that allows for two rotations of rear rod 3 with respect to the base: the first type of rotation with axis Oy, and the second type of rotation with axis Ox. Each of the first- and second-type rotations can be operated independently, and the two first- and second-type rotations can be combined. According to the particular embodiment shown in Figure 2, the actuation of the first- and second-type rotations, of axes Oy and Ox is carried out by the joint action of two parallel linear motors. Alternatively, the actuation of each of the first- and second-type rotations is performed by a rotary motor [18]. In addition, the arm of the robot is connected to the base by a revolute joint of axis Ox between lower rod 1 and the base, allowing for eliminating degrees of freedom, and in particular preventing the rotation of axis Oy between the lower rod 1 and the base so that the lower rod 1 remains horizontal at all times, in the extension of axis Ox.

The robotic arm allows for the creation of an RCM by creating a shift between the centre of rotation of a spherical mechanism whose actuators are located at a sufficient distance from the working area of a surgeon, and the centre of rotation O, located in the surgical intervention area. The occupation of the surgeon’s working space is thus limited.

The insertion–extraction mechanism comprises an elastic element such as a spring, a mechanism to transmit a motion of rotation from te triangular rod 1 to triangular rod 2, and a mechanism to transform this motion into a translational motion. This mechanism includes an actuator such as a rotary motor for exerting a torque on triangular rod 1. A motor associated with a brake controls this mechanism. When the brake is released, the spring pulls on the triangular rod and, via the coupling transmission, lifts the tool. The spring can be linear or torsion. The advantage of the latter is that it is more compact.

The transmission of motion from triangular rod 1 to triangular rod 2 can be achieved via a 4-bar mechanism as shown in Figure 2 with rod 7 or any other mechanism as a belt. The objective is to free up the space near the surgeon and reduce the torque to be produced by the motors placed at the base.

The translational motion of the endoscope is guided by a linear guide. A crank system mechanism can be connected to triangular rod 2. We can also use a rack and pinion system or the motion of the belt between axes E and F.

The kinematics proposed in the patent to achieve the rotation was optimised in [19,20]. It is a 2UPS-1U spherical parallel mechanism with an optimal position in the centre of the desired workspace.

### 3.2. Description of the Produced Prototype

Several parallel robot architectures exist to produce the first two rotational movements. For the prototype, a drive with rotary motors was preferred over linear actuators as presented in [18]. This solution is based on the agile eye created in [21], and was applied to ear and facial surgical applications in [18].

Figure 4 is the digital model of the robot being assembled. Two brushless motors (Beckhoff, AM8023—Servomotor 1.20 Nm) with gearboxes (Wittenstein, AG2400-+TP010S-MF2-70) were mounted onto the robot base and controlled via an EtherCat bus [22]. To perform the tests and in the absence of the translation mechanism that is fixed to the operating table, transport handles are used.

The insertion/extraction movement of the endoscope is achieved with the movement of a belt that is guided by four pulleys located on an inner parallelogram. The stroke of the endoscope is directly related to the height of this parallelogram minus the size of the carriage.

A motor associated with an encoder allows for the endoscope to move up and down in a controlled manner via a belt and a brake (Figure 5). This allows for the surgeon to precisely return to a saved camera posture after sharpening the endoscope tip. The movements are performed at slow speeds as the movements are only performed with visual control (Figure 6). It is a positional control, and no motion planning is possible, as the anatomy of the ear is not known and may change during the operation depending on the surgeon’s incisions.

If the power supply is cut off, the brake opens and releases the movement of the belt. Then, the torsion spring connected to the belt causes it to move. When the spring retracts, it lifts the endoscope. This movement is very fast but safe for the patient as it is along the insertion axis. As the tip of the endoscope has no roughness, it cannot tear off pieces of the ear or nose. There is no motion planning, as it is the energy of the spring that produces the motion. Unlike a 6R robot that would be used to carry the endoscope, the movement is like a reflex movement that can be performed very quickly upon the detection of an alarm.

The origin of the movements is defined by the high position of the endoscope. As this position is fixed, it is possible to adjust the position of the endoscope optics in relation to the centre of rotation of the mechanism. Initially, a straight optic is used. It is planned to add a small motor to allow for the rotation of the optics if it is tilted.

A constant load spring is used to store as much elastic energy as the potential gravitational energy associated with the movement of the endoscope and its acceleration to retract it. The weight of the endoscope and the linear rail is $M_{end}=0.75$ kg. The force provided by the constant load spring must be greater to produce the acceleration of the endoscope. We selected a spring as $F_{spring}=50$ N to provide acceleration of $56\;{m\cdot s}^{-2}$. This allows for the spring to raise the endoscope 10 cm in less than 0.06 s.

The motor must overcome this force to insert the endoscope into the ears or sinuses. Neglecting friction, the spring generates a torque on the motor shaft that is a function of the pulley radius:(1)$$\Gamma_{max}=F_{spring}R_{pulley}=50\times 0.00615=0.3\;N\cdot m$$

To transmit this torque, we chose an electromagnetic clutch that is able to transmit 1 Nm. The Maxon engine (EC-max 22 Ø22 mm) with its gearbox (Planetary Gearhead GP 22 C Ø22 mm) can generate a maximal torque of 2.9 N·m, which the clutch cannot transmit (86 011 04E). To simplify the wiring of this motor and the limit sensors, an EPOS4 Compact 24/1.5 EtherCAT drive was used.

When power is supplied to the clutch, the motor allows for the controlled lifting and descending of the endoscope. When the clutch is released, the spring causes the endoscope to rise rapidly. The device for detecting the patient’s awakening is not considered in this article. A simple emergency stop button operated by the surgeon or nurse can be used as a first step.

## 4. Conclusions

On the basis of the observation that patients may move or wake up during an operation, and that the surgeon must regularly clean the endoscope optics, a new robot architecture was presented. This solution integrates a mechanism that meets both these needs. A motor-controlled translation allows for the endoscope to be inserted and extracted for cleaning. An elastic system using torsion spring stores enough energy to eject the endoscope as soon as there is a sign of awakening. Due to the rapid movement along the insertion axis, there is no danger to the patient. Only the detection of the movement can be time-consuming, as there is no computer processing for the creation of this movement.

The robot is being manufactured, and its control system is being programmed. A displacement strategy was devised in [23] to allow for the movement of the suction tool to be tracked by markers while smoothing out the oscillations of the surgeon’s hand. Tests on synthetic anatomical structures are necessary to validate this approach, and new functions for the safety and cleaning of the optics.

## 5. Patents

This work is connected with patent WO2021058448A1 “Surgery assistance device” (Chablat Damien, Michel Guillaume, Bordure Philippe), and the ownership of Centre National de la Recherche Scientifique (CNRS), CHU de Nantes.

## Figures and Tables

**Figure 1 sensors-22-05175-f001:**
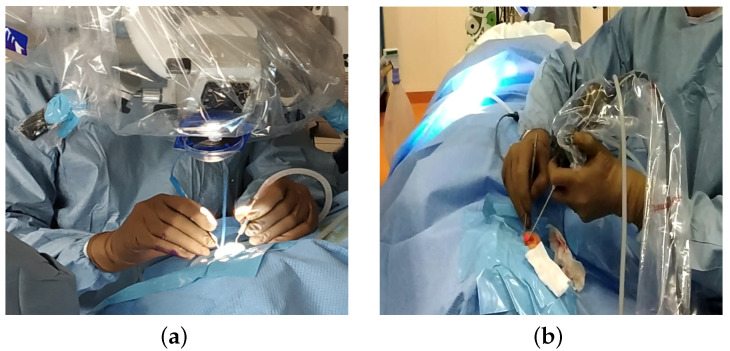
Comparison of the number of instruments that can be used simultaneously under the microscope and the endoscope. (**a**) Surgery under a microscope allowing for the use of both hands to operate. (**b**) Surgery under endoscopy, leaving only one hand for the operator to hold an instrument or suction.

**Figure 2 sensors-22-05175-f002:**
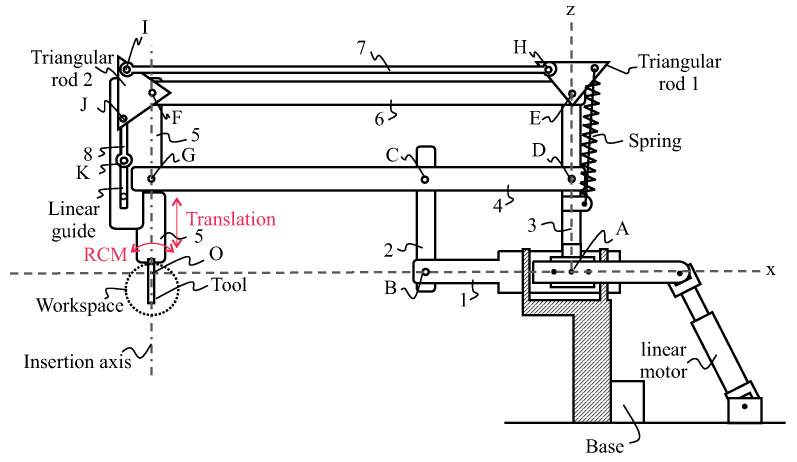
Candidate architecture of the robot with its security, where the numbers represent the names of the rods, and the letters the names of the rotation axes.

**Figure 3 sensors-22-05175-f003:**
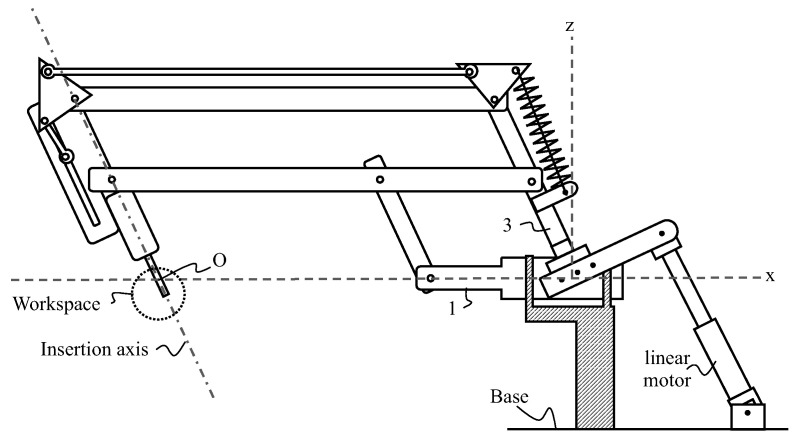
Candidate architecture in tilted position, where the numbers represent the names of the rods, and the letters the names of the rotation axes.

**Figure 4 sensors-22-05175-f004:**
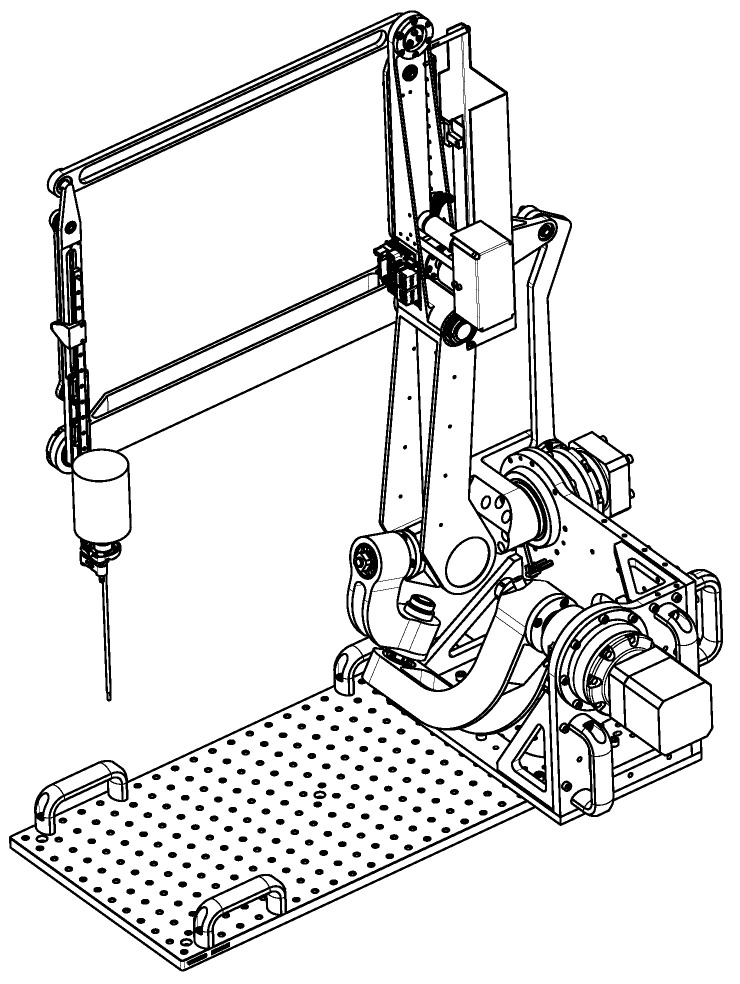
Prototype of the robot with an RCM.

**Figure 5 sensors-22-05175-f005:**
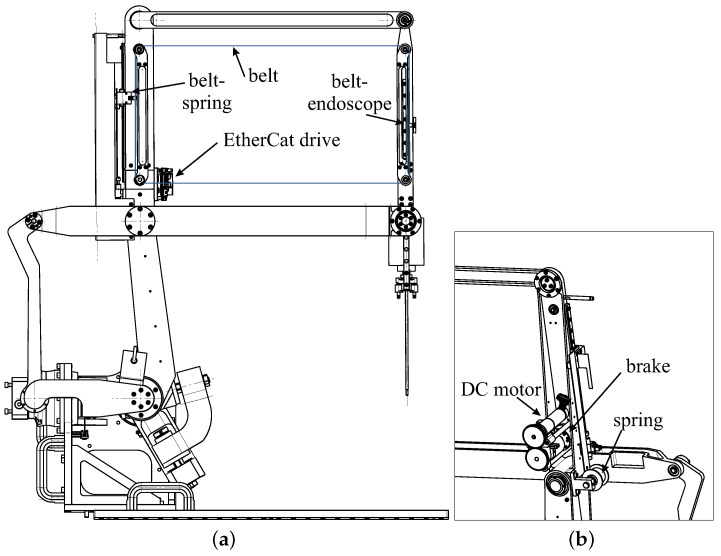
CAD modeling of the prototype (**a**) connection among DC motor, belt, brake, spring, and endoscope and (**b**) a zoom in on the arrangement of the spring, motor and brake.

**Figure 6 sensors-22-05175-f006:**
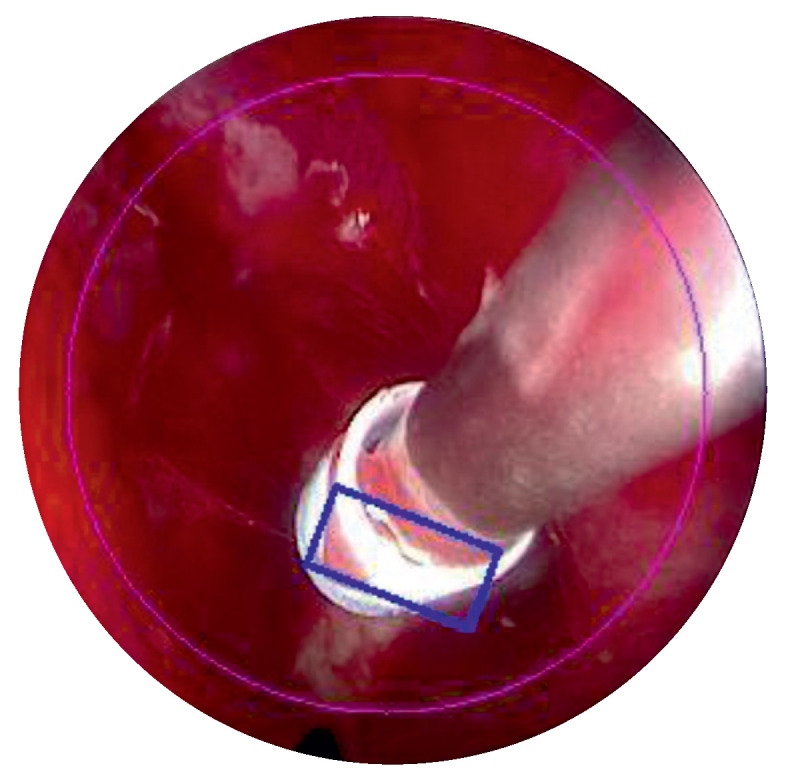
Object tracking by visual control.

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
