# Peer review of "A New Robotic Endoscope Holder for Ear and Sinus Surgery with an Integrated Safety Device"

_sensors, 2022, doi:10.3390/s22145175_

Round 1

Reviewer 1 Report

The authors explain the structure and configuration of a new robotic endoscope holder for ear and sinus surgery with an integrated safety device. Although the technology seems to be novel and patent-pending, as a research paper, I have the following concerns that need to be addressed:

- The paper discusses how this new mechanism reduces risks and increases safety; however, no proof or indication in terms of test or simulation results is reported to support this claim.

- The paper generally looks like a modified version of a patent languagte=ge in which the numbers have been removed, and the text has been rephrased. As a reader of Sensors, I expect to see some results supporting the idea, such as the implementation of the system, linkage config design (D-H notations or manipulability analysis) and etc.

Please add more technical content to the paper to make it a fit for Sensors.

Author Response

Response to the reviewers

We thank the reviewers for their critical assessment of our work. In the following, we address their concerns point by point.

Reviewer 1

The authors explain the structure and configuration of a new robotic endoscope holder for ear and sinus surgery with an integrated safety device. Although the technology seems to be novel and patent-pending, as a research paper, I have the following concerns that need to be addressed:

Reviewer Point P 1.1 — The paper discusses how this new mechanism reduces risks and increases safety; however, no proof or indication in terms of test or simulation results is reported to support this claim.

Reply: The authors cannot currently prove the improvement of patient safety. In the new version, the risks of injury have been developed from the medical literature.

There is little data on patient movement during ear surgery, although experience shows that this happens regularly (coughing, involuntary movements, early waking). However, a movement of a few millimeters against a rigid endoscope could have serious consequences. In a retrospective study of 100 consecutive patients undergoing otologic procedures [1], there was one instance where surgery was temporarily interrupted due to patient movement. But in a recent prospective study [2], a head motion was observed in 40 % of cases during ear surgeries. These values were close to those found in other prospective studies: a comparison of different sedation protocols (propofol- fentanyl and midazolam-fentanyl) revealed 30 to 35 % of movements during middle ear surgeries [6]; another comparative study found 23% (remifentanil-based anaesthesia) to 65% (propofol) of movements during surgery [4]. During robotic surgery, the maintenance of a deep neuromuscular blockade should be considered to increase safety by preventing patient movement [3]; but these drugs prevent the monitoring of the facial nerve, often required in otologic surgery.

Berges et al [2] measured these movements during otologic surgery: head motion in 40 % of cases with a maximum linear acceleration of 0.75 m/s2 and angular velocity of 12.50 degrees/s. In their opinion, these findings legitimize concerns that static endoscope holders represent a significant surgical risk, and demonstrate the need for a dynamic holder that could react to unintended head motion.

The data from the calculation gives a time of less than 0.06 s to remove the endoscope from

the patient’s head by 10 cm. An operator cannot do it any faster. On the current prototype under development, we do not have a tool to measure this time. See P1.3.

Reviewer Point P 1.2 — The paper generally looks like a modified version of a patent lan- guagte=ge in which the numbers have been removed, and the text has been rephrased. As a reader of Sensors, I expect to see some results supporting the idea, such as the implementation of the system, linkage config design (D-H notations or manipulability analysis) and etc.

Reply: The new version has simplified the presentation of the kinematics.

The arm of the robot comprises a deformable double parallelogram consisting of a first parallelogram comprising four rods 1, 2, 3, 4 and connected to a second parallelogram comprising four

rods 3, 4, 5, 6. The vertical rear rod 3 and the horizontal intermediate rod 4 are common to both parallelograms. The rods 1 to 6 are connected to each other by revolute joints A to G allowing rotations of axes parallel to the axis (Oy) and allowing in particular a rotation of axis (Oy) between the lower rod 1 and the rear rod 3. The insertion axis of the tool is oriented in the direction of the front rod 5, parallel to the rear rod 3 and the vertical intermediate rod 2.

The aim of the article is not to optimise the kinematics of the robot. For completeness, we have

added a reference on the optimisation of the spherical parallel robot used for the patent.

The kinematics proposed in the patent to achieve the rotation has been optimised in [5]. It is a 2UPS-1U spherical parallel mechanism with an optimal position in the centre of the desired workspace.

Reviewer Point P 1.3 — Please add more technical content to the paper to make it a fit for Sensors.

Reply: We have added a section on sizing the security system.

A constant load spring is used to store as much elastic energy as the potential gravitational energy associated with the movement of the endoscope and its acceleration to retract it. The weight of the endoscope and the linear rail is: Mend = 0.75 Kg. The force provided by the constant load spring has to be greater to produce acceleration of the endoscope. We select a spring as Fspring = 50 N to provide a acceleration of 56m.s−2. This allows the spring to raise the endoscope 10 cm in less than 0.06 s.

The motor must overcome this force to insert the endoscope into the ears or sinuses. The spring generates a torque on the motor shaft which is a function of the pulley radius,

Mmax = Fspring Rpulley = 50 0.00615 = 0.3N.m                                    (1)

To transmit this torque, we have chosen an electromagnetic clutch able to transmit 1 N.m. The Maxon engine (EC-max 22 Ø22 mm) with its gearbox (Planetary Gearhead GP 22 C Ø22 mm) can generate a maximum torque of 2.9 N.m which the clutch cannot transmit (86 011 04E). To simplify the wiring of this motor and the limit sensors, an EPOS4 Compact 24/1.5 EtherCAT drive is used. When power is supplied to the clutch, the motor allows controlled lifting and descending of the endoscope. When the clutch is de-energised, the spring causes the endoscope to rise rapidly. The device for detecting the patient’s awakening is not considered in this article. A simple emergency stop button operated by the surgeon or nurse can be used as a first step.

Reviewer 2 Report

The authors present a concept of an endoscope holder for ear and nose surgery. The main novelty of this device is the safety mechanism to quickly retract the endoscope form the surgical area when required. The concept is based on an author’s patent. The manuscript is well written and the concept is presented in an comprehensive manner.

Line 62: “static endoscope holders could represent a surgical risk” - The authors should back up this statement and provide data of  how often iatrogenic lesions of the middle ear or sinus structures occur in order to see if this safety mechanism is really necessary. For instance Maglio et al (reference 9) reported no such injuries in their series. So, please provide or two references stating how often this lesions occur.

Figure 2 – The description for what each point/ number represents should be provided in the subsidiary of the figure caption

Figure 3 – Same as figure 2

Line  133: “as un belt”- as an belt

There are 2 main limitation of this concept:

-        The lack of rotation movement mechanism for the endoscope, a very important feature to maximize the visualization area and access to remote zones of the working area

-        The safety mechanism has to be triggered manually and therefore is human depended.  Also, there is no clear description of how the safety spring mechanism is set off.

These limitation should be clearly stated. Perhaps, some options wo overpass them should be provided as well.

Author Response

Response to the reviewers

We thank the reviewers for their critical assessment of our work. In the following, we address their concerns point by point.

Reviewer 2

The authors present a concept of an endoscope holder for ear and nose surgery. The main novelty of this device is the safety mechanism to quickly retract the endoscope form the surgical area when required. The concept is based on an author’s patent. The manuscript is well written and the concept is presented in a comprehensive manner.

Reviewer Point P 2.1 — Line 62: “static endoscope holders could represent a surgical risk” - The authors should back up this statement and provide data of how often iatrogenic lesions of the middle ear or sinus structures occur in order to see if this safety mechanism is really necessary. For instance, Maglio et al (reference 9) reported no such injuries in their series. So, please provide or two references stating how often this lesions occurs.

Reply: The section was updated with new data:

There is little data on patient movement during ear surgery, although experience shows that this happens regularly (coughing, involuntary movements, early waking). However, a movement of a few millimeters against a rigid endoscope could have serious consequences. In a retrospective study of 100 consecutive patients undergoing otologic procedures [1], there was one instance where surgery was temporarily interrupted due to patient movement. But in a recent prospective study [2], a head motion was observed in 40 % of cases during ear surgeries. These values were close to those found in other prospective studies: a comparison of different sedation protocols (propofol- fentanyl and midazolam-fentanyl) revealed 30 to 35 % of movements during middle ear surgeries [6]; another comparative study found 23% (remifentanil-based anaesthesia) to 65% (propofol) of movements during surgery [4]. During robotic surgery, the maintenance of a deep neuromuscular blockade should be considered to increase safety by preventing patient movement [3]; but these drugs prevent the monitoring of the facial nerve, often required in otologic surgery.

Berges et al [2] measured these movements during otologic surgery: head motion in 40 % of cases with a maximum linear acceleration of 0.75 m/s2 and angular velocity of 12.50 degrees/s. In their opinion, these findings legitimize concerns that static endoscope holders represent a significant surgical risk, and demonstrate the need for a dynamic holder that could react to unintended head motion.

 Reviewer Point P 2.2 — Figure 2 – The description for what each point/ number represents should be provided in the subsidiary of the figure caption

Reply: The figure was updated as the text. We have added

where the numbers represent the names of the rods and the letters the names of the rotation axes.

Reviewer Point P 2.3  —  Figure 3 – Same as figure 2

Reply:  Same modifications.

Reviewer Point P 2.4  —  Line 133: “as un belt”- as an belt

Reply: The change is done. Thank you

There are two main limitations of this concept:

Reviewer Point P 2.5 — The lack of rotation movement mechanism for the endoscope, a very important feature to maximize the visualization area and access to remote zones of the working area.

Reply: The addition of rotation around the axis of the endoscope is not included in the current prototype. However, we can add a small motor at the end of the arm or leave it free to rotate with the surgeon’s hand. We do not have enough user data to define how this rotation axis is controlled at present. We have added:

Initially, a straight optic will be used. It is planned to add a small motor to allow the rotation of the optics if it is tilted.

Reviewer Point P 2.6 — The safety mechanism has to be triggered manually and therefore is human depended. Also, there is no clear description of how the safety spring mechanism is set off.

Reply: We have added the text:

When power is supplied to the clutch, the motor allows controlled lifting and descending of the endoscope. When the clutch is de-energised, the spring causes the endoscope to rise rapidly. The device for detecting the patient’s awakening is not considered in this article. A simple emergency stop button operated by the surgeon or nurse can be used as a first step.

These limitations should be clearly stated. Perhaps, some options to overpass them should be provided as well.

Reviewer 3 Report

The article presents in detail a patent showing a new robot for ear and sinus microsurgery. After analyzing in detail the problems in this type of surgery, a mechanism based on remote center motion is proposed for the insertion and rejection of the endoscope. 

Author Response

Reviewer 3

The article presents in detail a patent showing a new robot for ear and sinus microsurgery. After analyzing in detail the problems in this type of surgery, a mechanism based on remote center motion is proposed for the insertion and rejection of the endoscope.

Reply: Thank you for your comment

Round 2

Reviewer 1 Report

I would like to thank the reviewers for their effort to address my concerns. Although the paper has been revised, I need the authors to determine the contribution of the paper. To me:
1. The paper could be presented with a focus on the design of mechanism/linkage in which both kinematics (e.g. D-H notation) and dynamics (e.g. Lagrangian equations) of the mechanism should be mathematically addressed. Some examples are as follows:

- "A motor-controlled translation allows the endoscope to be inserted and extracted for cleaning." I suggest the authors support this with a path planning formulation (or presentation at least), which could be related to Figs 2 and 3.

- "Due to the rapid movement along the insertion axis, there is no danger to the patient. " How fast is this movement, and rapid compared to what? Can we have some numbers to sense the speed? Why the rapid movement causes no danger and makes the platform safe for the patient? Is there any test results or reference to refer to? 

2. What are the criteria to select the current mechanism over the other designs, any KPIs or surgical input?

3. Equation (1) is not correct to calculate the maximum torque of a motor. There are many other factors that need to be considered to calculate this value. although the motor works for this mechanism, the equation is only a very simplistic estimate.

4. Please respond to the above terms along with providing supporting documents for other claims in the paper to add significant scientific value to the manuscript. 

Author Response

Response to the reviewers

We thank the reviewer for their critical assessment of our work. In the following, we address their concerns point by point.

Reviewer 1

I would like to thank the reviewers for their effort to address my concerns. Although the paper has been revised, I need the authors to determine the contribution of the paper. To me:

Reviewer Point P 1.1 — The paper could be presented with a focus on the design of mecha- nism/linkage in which both kinematics (e.g. D-H notation) and dynamics (e.g. Lagrangian equa- tions) of the mechanism should be mathematically addressed. Some examples are as follows:

“A motor-controlled translation allows the endoscope to be inserted and extracted for cleaning.” I suggest the authors support this with a path planning formulation (or presentation at least), which could be related to Figs 2 and 3.

Reply: It is not necessary to present a table of DH parameters for just a double parallelogram. The mechanism that produces the rotation is presented in many publications. For the presented robot, as the motion will be performed at low speed, the dynamic model is not used. A well-calibrated PID produces sufficient trajectory tracking for vision control, as desired by the surgeons.

We have added this text for the motion control:

The movements are performed at slow speeds as the movements are with visual control only. It is a positional control and no motion planning is possible as the anatomy of the ear is not known and may change during the operation depending on the surgeon’s incisions.

Reviewer Point P 1.2 — ”Due to the rapid movement along the insertion axis, there is no danger to the patient. ” How fast is this movement, and rapid compared to what? Can we have some numbers to sense the speed? Why the rapid movement causes no danger and makes the platform safe for the patient? Is there any test results or reference to refer to?

Reply: We have added this text for the safety:

This movement is very fast but safe for the patient. The speed as the movement is along the insertion axis. As the tip of the andoscope has no roughness, it cannot tear off pieces of the ear or nose. There is no motion planning as it is the energy of the spring that produces the motion. Unlike a 6R robot that would be used to carry the endoscope, the movement is like a reflex movement that can be performed very quickly upon detection of an alarm.

The speed depends on the times because we have a constant acceleration. See definition of the

torques and times to eject.

Reviewer Point P 1.3 — What are the criteria to select the current mechanism over the other designs, any KPIs or surgical input?

Reply: Several other papers were written for the selection of the mechanism. See references [19], [20] and [21]. We have started the design by using questionnaire as shown in [19].

The main idea of any design is that the shape of the workspace should be regular and free from any kind of singularity. However, this is not the purpose of this article and the results of these studies have already been published.

Reviewer Point P 1.4 — Equation (1) is not correct to calculate the maximum torque of a motor. There are many other factors that need to be considered to calculate this value. Although the motor works for this mechanism, the equation is only a very simplistic estimate.

Reply: The formula does not introduce the maximum torque of the motor but the torque apply by the spring to the motor. I’m just add Neglecting friction, ....

Only one motor produces the up and down motion. No complex model has to be presented. It’s the

main advantage of the robot.

Reviewer Point P 1.5 — Please respond to the above terms along with providing supporting documents for other claims in the paper to add significant scientific value to the manuscript.

Reply: We have tried to answer the questions without changing the nature of this contribution, which is the presentation of an architecture including patient safety in ear and sinus operations. A thesis is starting for the evaluation of the prototype by its order and by tests on models provided by the CHU. The results are not yet available.